# HVAE: Hyperbolic Variational Autoencoder For Flexible Knowledge Transfer Across Multiple Domains

**Xiaolei Liu** [* 1] **Binfeng Wang** [* 1] **Kaixin Gao** [* 2] **Shaoshuai Li** [1]

## Abstract

Cross-domain recommendation (CDR) serves as a pivotal solution to data sparsity and cold-start problems by transferring knowledge across distinct domains. However, existing approaches predominately rely on Euclidean embedding spaces, which suffer from a fundamental **geometry-distribution mismatch**: real-world user-item interactions typically exhibit power-law distributions and latent hierarchical structures that flat Euclidean spaces cannot accurately represent without significant distortion. This geometric limitation not only compromises representation quality but, more critically, hinders the effective disentanglement of domain-invariant user preferences from domain-specific interests, limiting transferability in low-overlap scenarios. To bridge this gap, we introduce the **Mixed-Curvature Hyperbolic Variational Auto-Encoder (HVAE)**, a principled framework that unifies knowledge extraction and transfer within a hyperbolic manifold. By leveraging the exponential expansion capacity of hyperbolic geometry, HVAE naturally accommodates hierarchical data structures, enabling precise disentanglement of user intents without the need for strict domain overlap constraints. Furthermore, we propose a rigorous hyperbolic Wasserstein barycenter mechanism to align invariant distributions across heterogeneous domains. Extensive experiments on large-scale industrial and public datasets demonstrate that HVAE achieves superior performance, particularly in challenging scenarios with long-tail distributions and minimal domain overlap.

---

[*]Equal contribution [1]MYbank, Ant Group, Beijing, China [2]School of Mathematical Sciences, Ocean University, Qingdao, China. Correspondence to: Shaoshuai Li <llei14720@gmail.com>.

*Proceedings of the 43rd International Conference on Machine Learning*, Seoul, South Korea. PMLR 306, 2026. Copyright 2026 by the author(s).

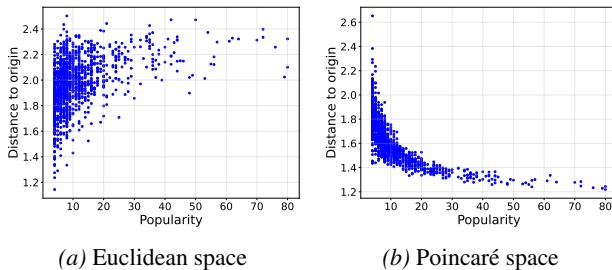

*(a) Euclidean space*  *(b) Poincaré space*

*Figure 1.* Distance to the origin $(0, 0)$ from each item vs its popularity in Euclidean and Poincaré spaces respectively. We take the music domain of Amazon dataset as an example and adopt a vanilla hyperbolic recommendation model, i.e., HGCF (Sun et al., 2021), to get the representations. (a) Euclidean version which is modified by HGCF; (b) Original hyperbolic version.

## 1. Introduction

Recommender Systems (RSs) have become indispensable in navigating the abundance of online content. While Collaborative Filtering (CF) (Sarwar et al., 2001) remains a dominant paradigm, it inherently struggles with *data sparsity* (Li et al., 2009) and the *cold-start* (Zhu et al., 2021b) problem. Cross-domain recommendation (CDR) offers a promising solution by transferring knowledge from rich source domains to target domains. The core efficacy of CDR hinges on two mechanisms: extracting reliable knowledge (representation) and sharing it effectively across domains (transfer). One dominant approach, grounded in matrix factorization, extracts intra-domain preferences and refines them via pairwise transfer modules (Yan et al., 2019; Li et al., 2019; Ma et al., 2019b; Zhao et al., 2019; Liu et al., 2020; Cui et al., 2020; Zhu et al., 2021a). An alternative strategy centers on aligning pre-trained domain-specific representations through explicit mapping functions (Man et al., 2017; Salah et al., 2021; Cao et al., 2022). Recent advancements have focused on disentanglement learning (Gao et al., 2025; Rong et al., 2025; Guo et al., 2023), aiming to separate a user's *domain-invariant* preferences from *domain-specific* behaviors.

Despite these advancements, we argue that existing methods overlook a foundational issue: the **Geometry-Distribution Mismatch**. Real-world interaction data in recommendation systems typically follows a power-law distribution, implying an underlying hierarchical structure (e.g., popular

items form the root,"tail" items form the leaves) (Gulcehre et al., 2018; Bronstein et al., 2017; Sala et al., 2018; Ravasz & Barabási, 2003) (see Appendix.A for empirical analysis). As shown in Fig.1, visualizing the learned representation differences between Euclidean and Poincaré spaces reveals that the distance to the origin in the vanilla hyperbolic model(Sun et al., 2021) increases exponentially with decreasing item popularity, aligning with hyperbolic volume expansion characteristics. Embedding such exponentially expanding hierarchies into a polynomial expanding Euclidean space inevitably leads to distortion, especially for "tail" users and items (Gulcehre et al., 2018; Sala et al., 2018). This distortion is fatal for CDR: if the underlying representation space is warped, any attempt to disentangle "invariant" from "specific" features becomes mathematically ill-posed and ineffective.

To address this, we propose the **Mixed-Curvature Hyperbolic Variational Auto-Encoder (HVAE)**. Instead of treating hyperbolic geometry as a mere plugin, we redesign the entire CDR pipeline to operate natively within the hyperbolic space. This choice is motivated by the fact that the exponential volume expansion of hyperbolic space allows HVAE to embed power-law data with minimal distortion, preserving the fidelity of long-tail user preferences.

Specifically, HVAE assigns adaptive mixed-curvature geometries to different domains to accommodate their heterogeneous densities. To facilitate robust knowledge transfer, we introduce a novel Hyperbolic Wasserstein Barycenter mechanism to effectively disentangle user representations into domain-invariant and domain-specific components via a variational framework. Additionally, to handle extreme sparsity where connectivity is scarce, we incorporate a self-training strategy with pseudo-labeling as a complementary enhancement.

Our main contributions are summarized as follows:

- **Geometric First Principles:** We identify the Euclidean geometry-distribution mismatch as a bottleneck in CDR and propose HVAE, the first framework to unify disentangled representation learning and knowledge transfer entirely within hyperbolic space.

- **Manifold-Aligned Transfer Mechanism:** We develop a computationally efficient method to approximate the Wasserstein Barycenter in hyperbolic space, enabling robust alignment of domain-invariant features across heterogeneous domains.

- **Scalability and Robustness:** We integrate an adaptive curvature mechanism and a self-training strategy, ensuring the model scales to multiple domains and remains robust in extreme cold-start settings. Extensive experiments on public and large-scale industrial

datasets confirm that HVAE achieves state-of-the-art performance, with significant gains in long-tail and cold-start settings.

## 2. Related work

**Multi-domain CDR.** Cross-domain recommendation(CDR) has seen significant advancements in addressing the cold-start and data-sparsity challenges. Early methods tackling cold-start issue, such as EM-CDR(Man et al., 2017) and SSC-DR(Kang et al., 2019), aligned independently modeled user interests across domains via mapping functions. More recent approaches, including TMCDR(Zhu et al., 2021b) and PUTPCDR(Zhu et al., 2022), leverage meta-networks trained on pre-trained user/item representations to dynamically generate personalized mappings. Variational autoencoder (VAE)-based methods, such as SA-VAE(Salah et al., 2021), VDEA(Liu et al., 2022), and DIDA-CDR(Zhu et al., 2023), further refine this line of research by disentangling latent user representations into discriminative and domain-invariant components. Meanwhile, methods tackling data sparsity have progressed from MLP-based transfer mechanisms like DDTCDR(Li & Tuzhilin, 2020) and CAT-ART(Zhao et al., 2022) to graph neural network (GNN)-based models such as GA-DTCDR(Zhu et al., 2020), CCDR (Xie et al., 2022),GA-MTCDR(Zhu et al., 2021a), Hero-Graph(Cui et al., 2020) and MSCDR (Li et al., 2023), which enhance cross-domain knowledge transfer via graph-based embeddings and attention-driven sharing modules. While these methods have demonstrated significant progress, they remain constrained by domain sensitivity and heavy reliance on extensive user overlap between domains. In contrast, our proposed approach offers a comprehensive solution that addresses both data-sparsity and cold-start issues in a unified framework. By eliminating sensitivity to the number of domains and user overlap constraints, our work establishes a foundation for building a universal cross-domain recommender system that is applicable across diverse scenarios.

**Domain Generalization**. Domain Generalization (DG) focuses on training models from multiple source domains to generalize well on unseen target domains, with significant progress made in the Computer Vision (CV) field. A popular branch to address domain shift is domain-invariant representation learning, which includes two main categories: explicit feature alignment(Tzeng et al., 2014; Motiian et al., 2017; Peng et al., 2019a; Wang et al., 2020; Zhou et al., 2020) and feature disentanglement like (Ding & Fu, 2017; Peng et al., 2019b; Ilse et al., 2020; Zhang et al., 2022). While DG methods have achieved remarkable success in CV tasks, their adaptation to multi-target cross-domain recommendation remains limited. Unlike CV tasks, CDR involves unique challenges, such as Euclidean representation distortion for user and item hierarchical embeddings. Our work

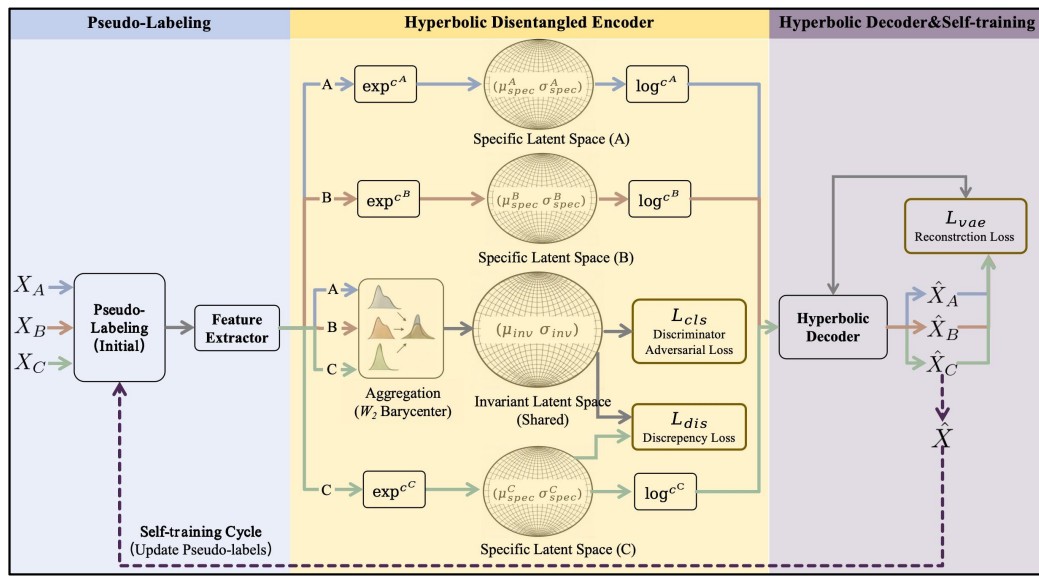

*Figure 2.* The overall framework of the proposed HVAE method. Leveraging a hyperbolic variational autoencoder, we disentangle user representations into domain-invariant and domain-specific components within hyperbolic space. Crucially, this representation is resilient to variations in the number of domains and the overlap among entities, making it highly adaptable to diverse cross-domain settings while preserving the hierarchical and structural properties of the data. The hyperbolic decoder reconstructs domain-specific interaction information for each user, ensuring fidelity to the latent representations generated by the encoder.

just bridges this gap by extending DG techniques to the multi-target CDR setting.

## 3. Methodology

In this section, we present the proposed **Mixed-Curvature Hyperbolic Variational Autoencoder (HVAE)**. Unlike previous approaches that treat embedding learning and domain adaptation as separate pipelines, HVAE unifies them into a single probabilistic framework on a Riemannian manifold. Fig.2 illustrates the overview of our proposed HVAE, which is the first attempt to unify data-sparsity as well as cold-start multi-domain CDR problems in non-Euclidean space. We first define the disentanglement mechanism within the hyperbolic space (Sec. 3.1.1). We then introduce a geometric alignment strategy based on the Hyperbolic Wasserstein Barycenter to enable robust knowledge transfer (Sec. 3.1.2). Finally, to handle scenarios with extreme data sparsity (e.g., $< 1\%$ overlap), we incorporate a cycle-consistent pseudo-labeling mechanism as an auxiliary data augmentation module (Sec. 3.3).

### 3.1. The HVAE Framework

We adopt the $n$-dimensional Poincaré ball model $\mathbb{B}^n_c$ as our Riemannian manifold, defined as $\mathbb{B}^n_c = \{x \in \mathbb{R}^n : c\|x\|^2 < 1\}$, where $c > 0$ denotes the curvature (See Appendix.B.2 for the related Riemannian manifold preliminary). The core premise of HVAE is to construct a variational inference framework on the hyperbolic manifold that disentangles a user $u$'s preferences $\mathbf{z}$ into two latent representations: a

**Domain-Invariant** component $z_{inv}$ and a **Domain-Specific** component $z_{spec}$. We achieve preference reconstruction and disentanglement by maximizing the Evidence Lower Bound (ELBO). The complete derivation of the hyperbolic ELBO is detailed in Appendix B.3.

#### 3.1.1. ADAPTIVE HYPERBOLIC ENCODER & DISENTANGLEMENT

Given the heterogeneity data distribution across domains, a fixed curvature limits model expressiveness. We therefore assign a learnable adaptive curvature $c_d$ to each domain $d \in \{A, B, C\}$, allowing the manifold to dynamically adapt to the intrinsic structure of each domain. Each domain is characterized by its user set $\mathcal{U}$, item set $\mathcal{V}$, and explicit binary user-item interaction matrix $\mathcal{R} \in \{0, 1\}^{|\mathcal{U}| \times |\mathcal{V}|}$. We do not impose strict conditions on the historical behaviors of users across domains, allowing for the possibility of no overlap between users in different domains.

**Hyperbolic Inference:** Specifically, given the raw binary interaction matrix $\mathcal{R}^d$ for domain $d$, the Encoder first maps the user preference representation to a hyperbolic embedding $u^d_g \in \mathbb{B}^n_{c_d}$. Being a crucial step prior to disentangled operations, we require two hyperbolic posterior distributions $q(z_{inv}|u, d)$ and $q(z_{spec}|u, d)$, aligned in hyperbolic VAE framework. Taking $z_{spec}$ as an example, its distribution parameters—the hyperbolic mean $\mu^d_{spec}$ and the tangent space variance $\sigma^d_{spec}$—are derived as:

$$\mu_{spec}^d = W_1 \otimes_{c_d} u_g^d \oplus_{c_d} b_1,$$
$$\sigma_{spec}^d = \text{Softplus}(W_2 \times \log_0^{c_d}(u_g^d) + b_2). \tag{1}$$

where $\otimes_{c_d}$ denotes Möbius matrix-vector multiplication under curvature $c_d$, $\{W_1, W_2, b_1, b_2\}$ are learnable parameters, and $\log_0^{c_d}(\cdot)$ is the logarithmic map projecting from the manifold to the tangent space. Then the posterior distribution $q(z_{spec}|u, d)$ can be obtained as $q(z_{spec}|u, d) = \mathcal{N}_{\mathbb{B}_{c_d}^n}(\mathbf{z} \mid \mu_{spec}^d, \sigma_{spec}^d)$. $\mathcal{N}_{\mathbb{B}_c^n}(\mathbf{z} \mid \boldsymbol{\mu}, \sigma^2)$ is the hyperbolic warpped normal distribution. See Appendix.B.4 for the derivation detail of probability density function and sampling strategy within hyperbolic space.

**Efficient Wasserstein Alignment:** To enforce geometric separation between $z_{inv}$ and $z_{spec}$, we aim to maximize the divergence between their distributions. However, computing the Wasserstein distance for continuous distributions in hyperbolic space typically requires sampling strategies with $O(n \log n)$ complexity, which prohibits scalability. We propose an analytical approximation that leverages the Gaussian properties in the tangent space, reducing complexity to $O(n)$. Specifically,

**Definition 3.1.** In hyperbolic space, the analytical approximated 2-Wasserstein distance of hyperbolic Gausssian distributions $\mathcal{W}p_1$ and $\mathcal{W}p_2$ is defined as:

$$W_2^2(\mathcal{W}p_{inv}d, \mathcal{W}p_{spec}^d) = \boldsymbol{dist}^2(\mu_{inv}^d, \mu_{spec}^d) +$$
$$((\sigma_{inv}^d)^{\frac{1}{2}} - (\sigma_{spec}^d)^{\frac{1}{2}})^2 \tag{2}$$

This formula combines the geodesic distance of means on the manifold with the Euclidean distance of standard deviations in the tangent space. Based on this, we construct the disentanglement loss $L_{dis}$ which need to be maximized to guide the disentanglement.

$$L_{dis} = \sum_{d=1}^n W_2^2(\mathcal{W}p_{inv}^d, \mathcal{W}p_{spec}^d) \tag{3}$$

Appendix B.5 provides the detailed definition.

### 3.1.2. INVARIANT INFORMATION AGGREGATION VIA BARYCENTER

Since the inferred "invariant" representation $z_{inv}$ may inherently vary across domains, we must aggregate information to obtain a globally consistent representation. Exploiting the geometry of the Wasserstein metric, we motivated by barycenter strategy to get the final and unique invariant mean $\hat{\mu}_{inv}$ and $\hat{\sigma}_{inv}$. Specifically, we first derive the unique invariant hyperbolic normal distribution as following strategy:

$$\mathcal{N}_{\mathbb{B}_c^n}(\hat{\mu}_{inv}, \hat{\sigma}_{inv}) = \arg\min_{\mu,\sigma} \sum_{d=1}^3$$
$$\lambda_d W_2^2\left(\mathcal{N}_{\mathbb{B}_c^n}(\mu, \sigma), \mathcal{N}_{\mathbb{B}_c^n}(\mu_{inv}'^d, \sigma_{inv}^d)\right) \tag{4}$$

then we have:

$$\hat{\mu}_{inv} = \arg\min_{\mu} \sum_{d=1}^3 \lambda_d \boldsymbol{dist}^2(\mu_{inv}'^d, \mu),$$
$$\hat{\sigma}_{inv} = (\sum_{d=1}^3 \lambda_d((\sigma_{inv}^d)^{\frac{1}{2}}))^2, \tag{5}$$

where $\lambda_d$ is calculate by the number of interactions in $d$-th domain. Specifically, let $num_d^u$ be the the number of interactions in $d$-th domain for one specific user, then

$$\lambda_d = \frac{e^{num_d^i}}{\sum_{j=1}^3 e^{num_j^i}}, i = 1, 2, 3. \tag{6}$$

Inspired by the work (Ungar, 2008), we involve a closed form expression to compute the average midpoint in the gyrovector spaces as:

$$m(x^{(1)}, ..., x^{(N)}; \lambda) = \frac{1}{2} \oplus \left( \sum_{i=1}^N \frac{\lambda_i \gamma_i}{\sum_{j=1}^N \lambda_j(\gamma_j - 1)} x^{(i)} \right) \tag{7}$$

with $\lambda = (\lambda_1, ..., \lambda_N)$ as the weights for each sample $x^{(i)}$ and $\gamma_i = \frac{2}{||x^{(i)}||^2}$. So $\hat{\mu}_{inv}$ can be obtained as

$$\hat{\mu}_{inv} = \frac{1}{2} \oplus \left( \sum_{i=1}^3 \frac{\lambda_i \gamma_i}{\sum_{j=1}^3 \lambda_j(\gamma_j - 1)} \mu_{inv}'^i \right), \tag{8}$$

where $\gamma_i = \frac{2}{||\mu_{inv}'^i||^2}$.

Having obtained the invariant posterior distribution, we employ an adversarial approach to enhance invariant learning. As domain shifts occur, invariant information remains constant, whereas specific information is closely tied to the domain index. We introduce a discriminator aimed at quickly identifying the domain index based on specific information. This is achieved by sampling latent variables from $q(z_{spec}|u, d)$ and using softmax prediction for multiclass domain classification. To prevent the discriminator from distinguishing the domain index, we sample latent variables from $q(z_{inv}|u)$, thereby confusing its judgment. Consequently, we incorporate the following auxiliary loss:

$$L_{cls} = -\sum_{d=1}^3 O_d \log(f_{cls}(z_{spec}^d)) - O_{inv} \log(f_{cls}(z_{inv})). \tag{9}$$

where $f_{cls}: \mathcal{M} \to \mathbb{R}^M$ represents the discriminator. $O_d = [0, \ldots, 1, \ldots, 0]$ is the domain index indicator and $O_{inv} = [1/3, 1/3, 1/3]$ if the number of domains is 3.

### 3.2. Hyperbolic Decoder

The decoding phase reconstructs user interactions from latent representations. We concatenate the aggregated invariant representation $Z_{inv}$ and the domain-specific representation $Z_{spec}^d$, then map them back to the manifold of domain $d$ via a fully connected layer $f_c$ to obtain the final user embedding $E^d = f_c(Z_{inv} \| Z_{spec}^d)$. $\|$ is a concat operator.

Then we compute the predicted score $s'(u_i, v_j)$ of user $i$ and item $j$ by Fermi-Dirac decoder, a hyperbolic generalization of the Sigmoid function, measuring the distance matching on the manifold:

$$s'(u_i, v_j) = \frac{1}{e^{(d_{\mathbb{B}}(E_i^d, V_j^d) - r)/t} + 1} \quad (10)$$

where $r$ and $t$ are hyperparameters. Thus, the reconstruction loss can be derived as follows:

$$\begin{aligned} \mathcal{L}_{vae} = &\sum_{i,j,d} -\mathcal{R}_{re(ij)}^d \log s'^d(u_i, v_j) \\ &+ \beta \sum_u KL(q(\mathbf{z}_{inv}) \| p(\mathbf{z}_{inv})) \\ &+ \beta \sum_{u,d} KL(q(\mathbf{z}_{spec}) \| p(\mathbf{z}_{spec})), \end{aligned} \quad (11)$$

.

### 3.3. Robust Training with Cycle-Pseudo-Labeling

While HVAE is geometrically robust, constructing stable manifold embeddings remains challenging for users with extreme sparsity or in cold-start scenarios using only sparse observed interactions. To address this, we integrate a self-training cycle that iteratively enhances the adjacency matrix via pseudo-labeling. To disambiguate variable definitions, we distinguish between the discrete binary interaction matrix $\mathcal{R} \in \{0,1\}^{|U| \times |V|}$ and the continuous similarity matrix $S \in [0,1]^{|U| \times |V|}$. In each training epoch $e$, we refine the adjacency matrix $S_{mod}^d$ for domain $d$ by blending observed data with model predictions:

$$S_{mod}^d = \alpha^e \cdot S_{obs}^d + (1 - \alpha^e) \cdot S_{pred}^d \quad (12)$$

where $S_{obs}^d$ is derived from the ground-truth observations $\mathcal{R}^d$. $S_{pred}^d$ is the current predicted similarity output by the HVAE decoder. $\alpha^e$ is a confidence weight that decays dynamically as training progresses.

This strategy allows $S_{mod}^d$ to rely on global priors during early training and progressively incorporate high-confidence pseudo-interactions "imagined" by HVAE. Experimental analysis in Industrial Online Performance section validates the robustness of this strategy in scenarios with only 1% overlap.

### 3.4. Optimization Objective

By merging all the loss functions mentioned above, we have the joint learning objective as:

$$L = L_{bpr} + \alpha_1 L_{vae} - \alpha_2 L_{dis} + \alpha_3 L_{cls}. \quad (13)$$

where $\alpha_1, \alpha_2, \alpha_3$ are hyperparameters balancing the terms. Regarding the optimizer, since the adaptive curvature $c_d$ causes the tangent space coordinate system to change dynamically, theoretical analysis and empirical results (see

Appendix B.6) demonstrate that the standard Adam optimizer is more stable and efficient than Riemannian Adam (Radam) in this specific setting.

### 3.5. Theoretical Analysis of HVAE

To theoretically justify the stability of the HVAE architecture in cross-domain knowledge transfer, we derive a formal generalization error bound for hypotheses defined on a Riemannian manifold. This analysis, grounded in the detailed proofs provided in Appendix B.7, elucidates how the negative curvature of the latent space acts as a geometric inductive bias to bound the target domain risk.

Consider the latent manifold $\mathbb{B}_c^n$ to be an $n$-dimensional Poincaré ball. We define a hypothesis space $\mathcal{H}$ consisting of $\phi$-Lipschitz continuous functions. For any $h \in \mathcal{H}$, the expected risk on the target domain $\mathcal{D}_t$ is denoted as $\epsilon_t(h) = \mathbb{E}_{(\mathbf{z},y) \sim \mathcal{D}_t}[L(h(\mathbf{z}), y)]$, and the empirical risk on the source domain $\mathcal{D}_s$ is $\hat{\epsilon}_s(h) = \frac{1}{m} \sum_{i=1}^m L(h(\mathbf{z}_i), y_i)$.

**Theorem.1.Manifold-Constrained Generalization Bound**
Following the derivation in Appendix B.7, for any $h \in \mathcal{H}$ and $\delta \in (0, 1)$, with probability at least $1 - \delta$, the following inequality holds:

$$\begin{aligned} \epsilon_t(h) \leq &\hat{\epsilon}_s(h) + 2\phi\hat{\mathfrak{R}}_m(\mathcal{H}) + 3\sqrt{\frac{\ln(2/\delta)}{2m}} \\ &+ \frac{1}{2} d_{\mathcal{H}\Delta\mathcal{H}}(\mathcal{D}_s, \mathcal{D}_t) + \lambda, \end{aligned} \quad (14)$$

where $\hat{\mathfrak{R}}_m(\mathcal{H})$ is the empirical Rademacher complexity, $d_{\mathcal{H}\Delta\mathcal{H}}$ is the $\mathcal{H}\Delta\mathcal{H}$-distance representing the domain divergence, and $\lambda$ is the minimum combined risk of the ideal joint hypothesis. Theorem 1 highlights the fundamental advantages of the HVAE architecture through the lens of *curvature-driven complexity control*:

- **Capacity Regulation via Negative Curvature:** In the Poincaré ball, the volume of a geodesic ball $\mathrm{Vol}(\mathbb{B}_R)$ grows exponentially with $R\sqrt{c}$. While this provides high representation capacity for hierarchical data, the $\hat{\mathfrak{R}}_m(\mathcal{H})$ term is implicitly regularized by the Riemannian metric. Maintaining $\phi$-Lipschitz continuity on a negatively curved manifold is more restrictive than in Euclidean space, effectively compressing the hypothesis space and leading to a tighter generalization gap.

- **Divergence Minimization:** As proved in Appendix B.7, the fidelity of hyperbolic geometry to hierarchical topologies significantly reduces the $d_{\mathcal{H}\Delta\mathcal{H}}$ divergence. By aligning multiple domains within a unified Poincaré ball, HVAE minimizes the discrepancy between $\mathcal{D}_s$ and $\mathcal{D}_t$, ensuring that the transfer error remains bounded even with limited target samples $m$.

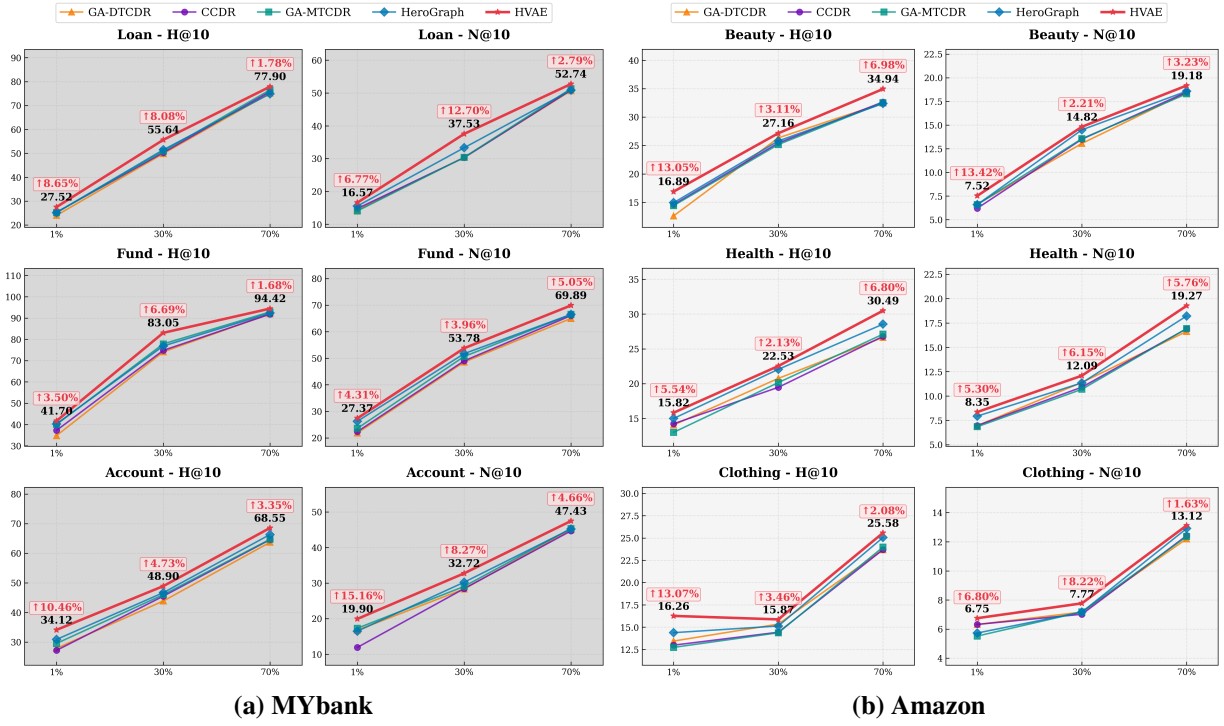

*Figure 3.* Model performance (%) under data sparsity scenarios.

## 4. Experiments

**Experiment Protocol**. To address our research inquiries, we executed empirical investigations utilizing publicly Amazon datasets[1] and a real-world large-scale financial dataset collected from MYbank of Ant Group[2]. Details are presented in Table.1. Following common practice in previous RS literatures(Zhu et al., 2021a; Li & Tuzhilin, 2020; Zhu et al., 2019), we employ widely accepted $top$-$N$ metrics normalized discounted cumulative gain (N@10) and hit rate (H@10) as evaluation metrics for CDR scenarios. Higher values means better performance for all the metrics. Besides, we selected four established benchmarks: GA-DTCDR(Zhu et al., 2020) and CCDR(Xie et al., 2022) generally used for dual-domain cross-domain recommendation tasks, and GA-MTCDR(Zhu et al., 2021a) and HeroGraph(Cui et al., 2020) for multi-domain cross-domain recommendation scenarios. We choose these baselines because they have been repeatedly validated in industrial scenarios, demonstrating strong performance and stability.

**Parameter Settings**. To ensure a fair comparison across methods, we maintained consistent hyperparameter settings for all experiments, which were conducted on A100 GPU using Pytorch, with results averaged over 5 runs; the reported hyperparameters represent the best outcomes from extensive random searches. We also examined the impact of varying

user overlap ratios—$1\%, 30\%, 70\%$—on the dataset distribution, with lower percentages indicating less overlap and higher percentages indicating greater overlap. For method-specific hyperparameters, we tuned the VAE factor $\alpha_1$, the distance factor $\alpha_2$, and the discriminator factor $\alpha_3$ in Eq. 13 within $\{0.00001, 0.0001, 0.001, 0.01, 0.1, 1\}$.

*Table 1.* Statistics of the dataset.

| Task | Amazon | | | MYbank | | |
|---|---|---|---|---|---|---|
| Domain | Beauty | Health | Clothing | Loan | Fund | Account |
| #User | 36332 | 36778 | 20729 | 947139 | 622198 | 293276 |
| #Item | 17973 | 22536 | 11746 | 1921 | 717 | 2770 |
| #Avg.I/User | 15.26 | 11.50 | 13.34 | 2190.82 | 775.13 | 307.5 |
| #Avg.U/Item | 7.55 | 7.05 | 7.56 | 4.44 | 0.89 | 2.09 |
| #Actions | 274182 | 259248 | 156649 | 4208565 | 555765 | 851762 |
| #Sparsity | 99.96% | 99.97% | 99.94% | 99.77% | 99.88% | 99.90% |

### 4.1. Performance Analysis

**How does HVAE perform on data-sparsity scenarios?** A critical limitation of existing Cross-Domain Recommendation (CDR) methods is their reliance on high user overlap ratios (often assuming $> 50\%$) to align feature spaces. As illustrated in Fig. 3 (where red arrows indicate the performance gains of HVAE over the second-best baseline), HVAE consistently achieves state-of-the-art results across all user overlap ratios ($1\%, 30\%, 70\%$), datasets (MYbank, Amazon), and evaluation metrics ($H@10, N@10$).

Critically, the performance margin between HVAE and the strongest baseline increases monotonically as overlap decreases, revealing its structural robustness under extreme data scarcity. Under conditions of extreme sparsity ($1\%$

---

[1]http://jmcauley.ucsd.edu/data/amazon/

[2]https://www.antgroup.com/en

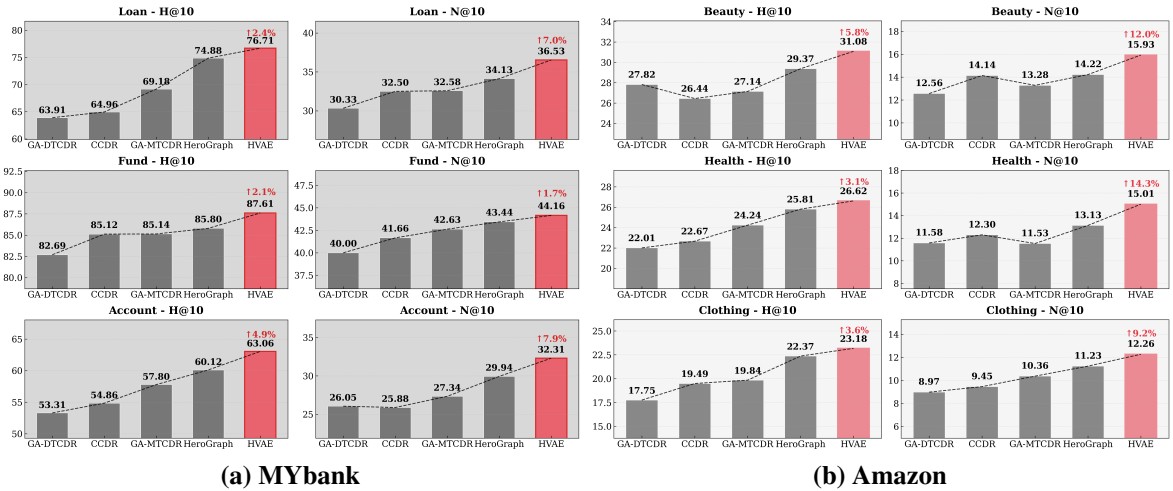

*Figure 4.* Model performance (%) under cold-start scenarios.

overlap), HVAE yields significant improvements of 3.5%–10.46% ($H$@10) and 4.31%–15.16% ($N$@10) across all domains in the MYbank dataset—margins that represent substantial gains for large-scale industrial applications. Similarly, on the Amazon dataset, HVAE achieves improvements of 5.54%–13.07% ($H$@10) and 5.3%–13.42% ($N$@10), demonstrating exceptional efficacy for long-tail user modeling. At moderate sparsity (30% overlap), HVAE maintains its superiority, on the MYbank dataset, HVAE achieves average gains of 6.5% ($H$@10) and 8.31% ($N$@10), while on the Amazon dataset, it yields average improvements of 2.9% ($H$@10) and 5.53% ($N$@10). Even under high-overlap conditions (70%), where anchor-rich settings theoretically favor conventional alignment approaches, HVAE retains a measurable lead. The superiority of HVAE stems from its geometric properties. Euclidean models suffer from the "crowding problem" where tail users (who have few interactions) are squeezed near the origin, making them indistinguishable during transfer. HVAE's hyperbolic geometry provides exponential volume near the boundary, allowing it to preserve the hierarchical distinction of these sparse users even when direct overlapping signals are minimal.

**How does HVAE perform on cold-start scenarios?** We further evaluate the performance of our model in cold-start scenarios, where training users are randomly selected, and test users are entirely new to the target domain. Specifically, to assess the cold-start performance of a user $u_1$ in domain A, we exclude $u_1$'s interaction data from domain A in the training set. The results, shown in Fig. 4, demonstrate that our HVAE model outperforms other methods across all evaluation metrics, with an average improvement of 3.13% and 5.53% over the second-best method in H@10 and N@10, respectively on MYbank. Similar findings are obtained in experiments on the Amazon dataset. The superior performance of HVAE in cold-start scenarios is attributed to its

ability to decouple information and model user-item interactions in non-Euclidean space, which, combined with the utilization of pseudo-label techniques, enables HVAE to effectively leverage knowledge from other domains and iteratively refine the cold-start user representations. These capabilities lead to more accurate and robust recommendations for users in the target domain with limited prior data.

### 4.2. Ablation Study

To investigate the contributions of HVAE's key components, we conducted an extensive ablation study by systematically removing or modifying critical modules. The following variants were examined:**w/o-Hypolic**: replacing hyperbolic space with Euclidean space for representation learning; **w/o-Adaptive**: using fixed curvatures for each domain instead of learnable ones; **w/o-VAE**: replacing the variational autoencoder with a deterministic autoencoder; **w/o-Pseudo-label**: removing the pseudo-labeling and cycle-training mechanisms; **w/o-Distance**: disabling the distance-based loss by setting $\alpha_2 = 0$; **w/o-Classify**: eliminating the classification loss by setting $\alpha_3 = 0$. Fig. 5 presents the results. Notable observations include: **Impact of Hyperbolic Geometry:** Substituting hyperbolic embeddings with their Euclidean counterparts results in the most pronounced performance degradation. This observation empirically validates our primary motivation: the intrinsic necessity of hyperbolic geometry for robust hierarchical feature extraction and cross-domain knowledge transfer. **Importance of Adaptive Curvature:** Utilizing fixed curvatures rather than learnable ones leads to a substantial decline in efficacy. This underscores the objective necessity of assigning domain-specific, learnable curvatures to effectively model the intrinsic data heterogeneity inherent in different domains. **Variational vs. Deterministic Modeling:** Replacing the variational

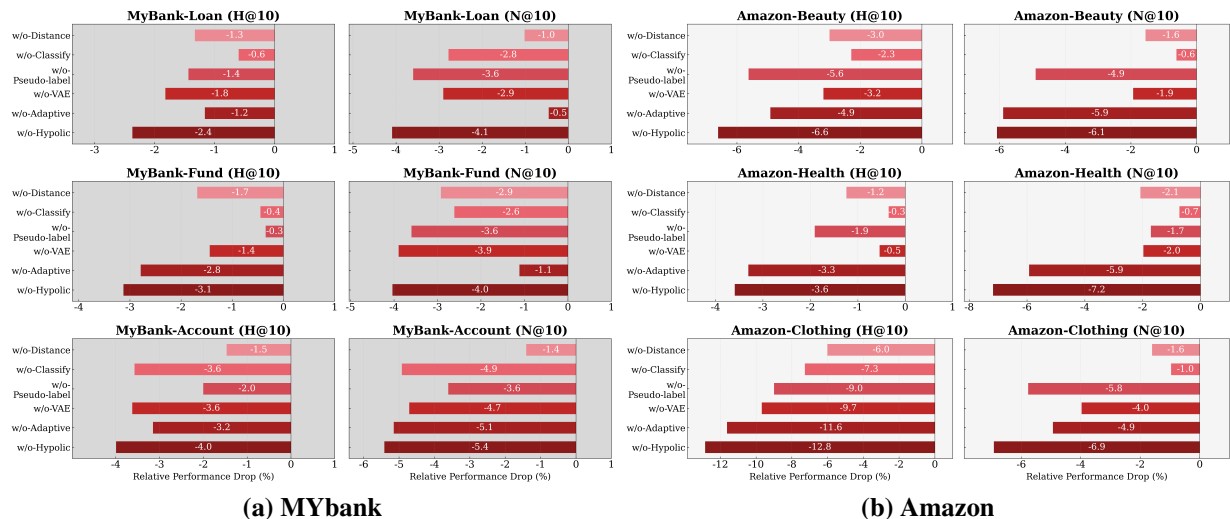

*Figure 5.* Relative Performance Drop of HVAE Components.

autoencoder framework with a deterministic autoencoder yields a noticeable drop in performance, suggesting that the probabilistic latent space is crucial for capturing uncertainty and ensuring invariant knowledge transfer. **Besides**, the pseudo-labeling strategy also provides a substantial performance boost to the HVAE. In conclusion, these findings demonstrate that each module is indispensable; their synergistic integration underpins the overall high-performance behavior of the HVAE model.

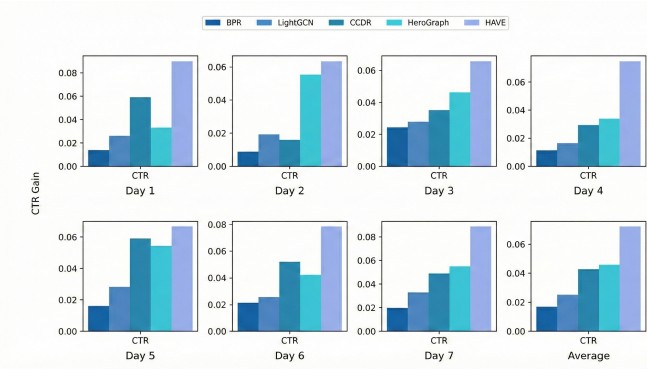

*Figure 6.* Online results for A/B test within 7 days.

### 4.3. Industrial Online Testing

To evaluate the effectiveness of HVAE in real-world online scenarios, we conduct an online A/B test on the Ant Group recommendation platform. Table. 3 (See AppendixC) presents five domains of real-world scenarios with a low overlap usage ratio of 1% across all domains. Note that during the online experiments, we additionally included BPR and LightGCN, which are classic single-domain recommendation methods that have been repeatedly used in the industry. Fig. 6 illustrates the online CTR (a critical metric for evaluating user engagement in online advertising, recommendation systems, and e-commerce) results over a period

of 7 days, showing that HVAE outperformed all other models significantly, i.e. 7.453% improvement on average. The results prove HVAE's effectiveness in large-scale, challenging environments with multiple domains (at least five), low user overlap (approximately 1%), large user bases (nearly ten million per domain) and rigorously real-time inference constraints.

## 5. Conclusion

In this paper, for the sake of developing a method that addresses both long-standing data scarcity and cold-start RSs issues from a fresh perspective, we propose an innovative adaptive mixed-curvature HVAE framework that enables us to jointly learn and transfer reliable knowledge across multiple domains in a hyperbolic space. Besides, we offer theoretical understandings to demonstrate the reasoning behind our proposed approach. In essence, our HVAE can be viewed as a more versatile extension of the Euclidean VAE framework. Comprehensive testing and ablation studies have validated the remarkable effectiveness of the proposed method across various evaluation metrics.

## Impact Statement

This paper presents work whose goal is to advance the field of Machine Learning. There are many potential societal consequences of our work, none which we feel must be specifically highlighted here.

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

## A. Dataset Distribution

As discussions in many works (Ravasz & Barabási, 2003; Wang et al., 2015; 2018; Ma et al., 2019a; Li et al., 2020), user-item interactions of recommendation systems generally present the intrinsic power-law distribution, which means that a majority of user/item have very few interactions and a few users/items have a huge number of interactions. We present the distribution of number of interactions for both user and item in the following Fig. 7 and Fig. 8 respectively. Power-law distribution is obvious in both the Amazon and MYbank datasets, where a majority of users interact with items very few times, meanwhile, most items are with few clicks. Such power-law phenomenon can be approximately described as the 'hierarchical' structure according to the number of interactions slicing. Users (or items) with similar number of interactions are at the same level. For example, in the first level, users (or items) have the highest number of active clicks (for items, referring as the highest number of passive clicks by users), while those in the last level have the lowest active clicks. The number of users or items grows exponentially as the number of interactions decrease. In other words, the number of users or items grows exponentially with the increasing of distance from the hierarchy origin.

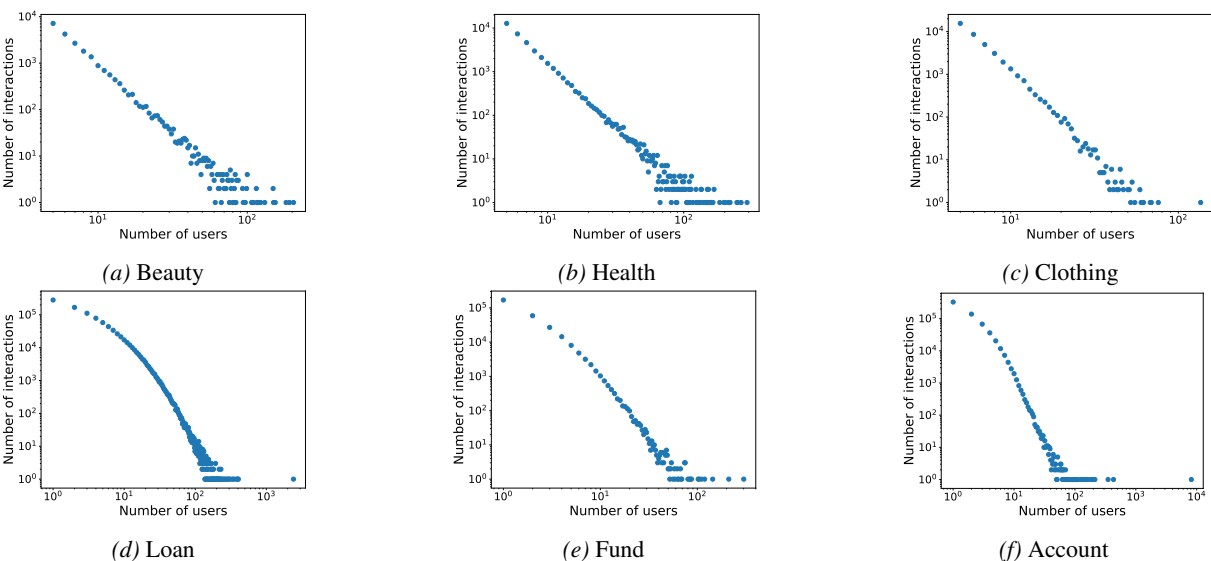

*Figure 7.* Clicking distributions of users in each of the datasets. The $x$-axis presents the number of such users, and the $y$-axis shows the number of interactions associated with a user.

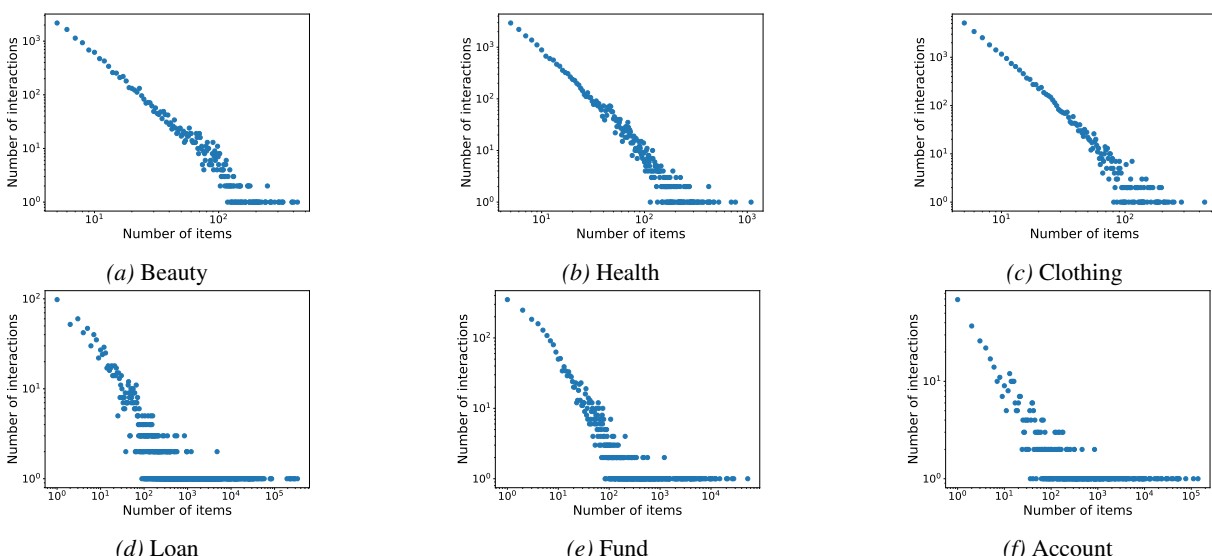

*Figure 8.* Clicking distributions of items in each of the datasets. The $x$-axis presents the number of items, and the $y$-axis shows the number of interactions associated with an item.

## B. Methodology Supplementation

### B.1. Notation Table

Table 2 summarizes the key notations used throughout the paper.

*Table 2.* Summary of Notations

| Symbol | Definition |
|--------|------------|
| $\mathcal{M}$ | Riemannian manifold (general) |
| $\mathbb{B}_c^n$ | $n$-dimensional Poincaré ball with curvature $c$ |
| $\mathcal{T}_x\mathbb{B}_c^n$ | Tangent space at point $x \in \mathbb{B}_c^n$ |
| $g^{\mathbb{B}}$ | Riemannian metric tensor of the Poincaré ball |
| $\exp_x^c / \log_x^c$ | Exponential / Logarithmic map at $x$ with curvature $c$ |
| $\oplus_c$ | Möbius addition with curvature $c$ |
| $d \in \{A, B, C\}$ | Domain index |
| $\mathcal{U}, \mathcal{V}$ | Sets of users and items |
| $\mathcal{R}^d$ | Binary interaction matrix for domain $d$ ($\{0, 1\}$) |
| $S^d$ | Continuous similarity matrix for domain $d$ ($\in [0, 1]$) |
| $c_d$ | Learnable adaptive curvature for domain $d$ |
| $u_g^d, v_g^d$ | Hyperbolic embeddings of user/item in domain $d$ |
| $z_{inv}$ | Domain-invariant latent representation (Hyperbolic) |
| $z_{spec}^d$ | Domain-specific latent representation (Hyperbolic) |
| $\mu, \sigma$ | Mean (on manifold) and Variance (in tangent space) |
| $\lambda_d$ | Weight for domain $d$ in Barycenter aggregation |
| $L_{vae}, L_{dis}$ | VAE reconstruction loss and Disentanglement loss |
| $L_{cls}, L_{bpr}$ | Adversarial classification loss and BPR ranking loss |

### B.2. Riemannian Manifold Preliminary

Here, we provide the complete geometric operations required for the HVAE Encoder and Decoder.

**Metric and Distance.** The Riemannian metric tensor $g_x^{\mathbb{B}}$ at point $x \in \mathbb{B}_c^n$ is conformal to the Euclidean metric $g^{\mathbb{E}}$:

$$g_x^{\mathbb{B}} = (\lambda_x^c)^2 g^{\mathbb{E}}, \quad \text{where } \lambda_x^c = \frac{2}{1 - c\|x\|^2} \tag{15}$$

$\lambda_x^c$ is the conformal factor. The induced geodesic distance between two points $x, y \in \mathbb{B}_c^n$ is:

$$d_{\mathbb{B}_c^n}(x, y) = \frac{2}{\sqrt{c}}\tanh^{-1}(\sqrt{c}\| - x \oplus_c y\|) \tag{16}$$

**Mapping Operations.** To perform operations (such as convolution or sampling) in hyperbolic space, we utilize the tangent space $\mathcal{T}_x\mathbb{B}_c^n \cong \mathbb{R}^n$.

- **Logarithmic Map** ($\mathbb{B}_c^n \to \mathcal{T}_x\mathbb{B}_c^n$): Projects a point $y$ on the manifold to the tangent space at $x$.

$$\log_x^c(y) = \frac{2}{\sqrt{c}\lambda_x^c}\tanh^{-1}(\sqrt{c}\| - x \oplus_c y\|)\frac{-x \oplus_c y}{\| - x \oplus_c y\|} \tag{17}$$

- **Exponential Map** ($\mathcal{T}_x\mathbb{B}_c^n \to \mathbb{B}_c^n$): Maps a vector $v$ from the tangent space back to the manifold.

$$\exp_x^c(v) = x \oplus_c \left(\tanh\left(\frac{\sqrt{c}\lambda_x^c\|v\|}{2}\right)\frac{v}{\sqrt{c}\|v\|}\right) \tag{18}$$

.

### B.3. Derivation of Hyperbolic ELBO

Here, we provide the complete derivation of the Evidence Lower Bound (ELBO). Given the observed user interactions $u$ and the domain index $d$, our goal is to maximize the log-likelihood $\log p(u)$. We postulate that the user's latent preference

consists of a domain-invariant component $z_{inv}$ and a domain-specific component $z_{spec}$. According to variational inference theory, we introduce two approximate posterior distributions, $q_\phi(z_{inv}|u, d)$ and $q_\phi(z_{spec}|u, d)$, to approximate the true posteriors.

Applying Jensen's Inequality, the lower bound of $\log p(u)$ is derived as follows:

$$
\begin{aligned}
\log p(u) &= \log \iint p(u, z_{inv}, z_{spec}, d) \, dz_{inv} \, dz_{spec} \\
&= \log \iint \frac{p(u|z_{inv}, z_{spec}, d)p(z_{inv})p(z_{spec}|d)}{q_\phi(z_{inv}|u, d)q_\phi(z_{spec}|u, d)} q_\phi(z_{inv}|u, d)q_\phi(z_{spec}|u, d) \, dz_{inv} \, dz_{spec} \\
&\geq \mathbb{E}_{q_\phi}\left[\log p(u|z_{inv}, z_{spec}, d) + \log \frac{p(z_{inv})}{q_\phi(z_{inv}|u, d)} + \log \frac{p(z_{spec}|d)}{q_\phi(z_{spec}|u, d)}\right] \\
&= \underbrace{\mathbb{E}_{q_\phi}[\log p(u|z_{inv}, z_{spec}, d)]}_{\text{Reconstruction Term}} - \underbrace{\text{KL}(q_\phi(z_{inv}|u, d)\|p(z_{inv}))}_{\text{Invariant Regularization}} - \underbrace{\text{KL}(q_\phi(z_{spec}|u, d)\|p(z_{spec}|d))}_{\text{Specific Regularization}}
\end{aligned}
\tag{19}
$$

### B.4. Wrapped Normal Distribution

To parameterize the probability distributions $q(z|u)$ on the Poincaré ball $\mathbb{B}_c^n$, we utilize the **Wrapped Normal Distribution** ($\mathcal{N}_\mathbb{B}$). This distribution is defined by projecting a Gaussian distribution from the Euclidean tangent space onto the hyperbolic manifold.

**1. Sampling via Reparameterization.** A random variable $z \in \mathbb{B}_c^n$ following $\mathcal{N}_\mathbb{B}(\mu, \Sigma)$ can be generated through the following steps:

1. Sample a standard Gaussian noise $v \sim \mathcal{N}(0, \Sigma)$ in the tangent space at the origin $\mathcal{T}_0\mathbb{B}_c^n$ (isomorphic to $\mathbb{R}^n$).

2. Transport the vector $v$ to the tangent space at the mean $\mathcal{T}_\mu\mathbb{B}_c^n$ using Parallel Transport ($PT_{0\to\mu}$).

3. Project the tangent vector onto the manifold using the Exponential Map ($\exp_\mu^c$).

Formally:
$$
z = \exp_\mu^c(PT_{0\to\mu}(v)), \quad v \sim \mathcal{N}(0, \Sigma)
\tag{20}
$$

This process is differentiable, supporting the reparameterization trick required for gradient-based optimization of HVAE parameters.

**2. Probability Density Function.** To compute the KL divergence in Appendix B.3, the explicit probability density function (PDF) of $z$ is required. Mapping from a flat tangent space to a curved manifold induces volume changes, which necessitates a Jacobian determinant term (Volume Correction Factor).

For $z \in \mathbb{B}_c^n$, the PDF is defined as:

$$
\mathcal{N}_{\mathbb{B}_c^n}(z|\mu, \Sigma) = \mathcal{N}(u|0, \Sigma) \cdot \underbrace{\left(\frac{\sqrt{c}d_\mathbb{B}(z, \mu)}{\sinh(\sqrt{c}d_\mathbb{B}(z, \mu))}\right)^{n-1}}_{\text{Volume Correction Factor}}
\tag{21}
$$

Where:

- $u = \log_\mu^c(z)$ is the vector corresponding to $z$ in the tangent space $\mathcal{T}_\mu\mathbb{B}_c^n$.

- $\mathcal{N}(\cdot)$ is the standard Euclidean Gaussian density function.

- $d_\mathbb{B}(z, \mu)$ is the geodesic distance between $z$ and $\mu$.

- The correction factor $\left(\frac{r}{\sinh r}\right)^{n-1}$ accounts for the exponential volume expansion characteristic of hyperbolic space.

we set the priors for the disentangled representations as:

- **Invariant Prior:** $p(z_{inv}) = \mathcal{N}_{\mathbb{B}}(0, I)$ (Standard Hyperbolic Normal).

- **Specific Prior:** $p(z_{spec}|d) = \mathcal{N}_{\mathbb{B}}(\mu_d, \sigma_d)$ (Parameterized Hyperbolic Normal with learnable $\mu_d, \sigma_d$).

### B.5. Analytical Approximation of Hyperbolic Wasserstein Distance

In Section 3.1.1, we introduced a computationally efficient approximation for the Wasserstein distance to calculate $L_{dis}$. Here we justify this approximation.

The squared 2-Wasserstein distance between two Euclidean Gaussian distributions $\mathcal{N}(\mu_1, \Sigma_1)$ and $\mathcal{N}(\mu_2, \Sigma_2)$ is defined as:

$$W_2^2 = \|\mu_1 - \mu_2\|_2^2 + \text{trace}(\Sigma_1 + \Sigma_2 - 2(\Sigma_1^{1/2}\Sigma_2\Sigma_1^{1/2})^{1/2}) \tag{22}$$

Assuming diagonal covariance matrices (a common practice in VAEs), this simplifies to:

$$W_2^2 = \|\mu_1 - \mu_2\|_2^2 + \|\Sigma_1^{1/2} - \Sigma_2^{1/2}\|_F^2 \tag{23}$$

In Hyperbolic Space, a closed-form solution for Wasserstein distance between Wrapped Normal distributions does not exist. Standard approaches require numerical integration or sampling, scaling at $O(n \log n)$ or worse. To achieve $O(n)$ complexity for industrial scalability, we approximate the distance by decoupling the transport cost into:

1. **Transport of Mean:** Measured by the geodesic distance on the manifold $d_{\mathbb{B}_c^n}(\mu_1, \mu_2)$.

2. **Transport of Mass (Shape):** Measured by the Euclidean distance of the tangent space standard deviations, exploiting the fact that the "shape" of the distribution is defined in the local tangent space.

Thus, we propose the approximation used in the main text:

$$W_2^2(\mathcal{W}_{p1}, \mathcal{W}_{p2}) \approx \underbrace{d_{\mathbb{B}_c^n}^2(\mu_1, \mu_2)}_{\text{Manifold Mean Distance}} + \underbrace{\|\sigma_1^{1/2} - \sigma_2^{1/2}\|_2^2}_{\text{Tangent Shape Distance}} \tag{24}$$

This approximation preserves the metric properties required to force the separation of $z_{inv}$ and $z_{spec}$ while fitting within the computational budget of large-scale recommendation systems.

### B.6. Optimizer Analysis: Adam vs. Radam

In Section 3.4, we chose Adam over Riemannian Adam (Radam). The theoretical justification is as follows:

Radam relies on the concept of Parallel Transport to move momentum vectors across tangent spaces as parameters update. This assumes a fixed manifold structure. However, our HVAE employs Adaptive Curvature ($c_d$), meaning the manifold structure itself ($\mathbb{B}_{c_d}^n$) changes at every gradient step.

- **Radam's limitation:** Updating $c_d$ changes the metric tensor globally. A momentum vector $v_t$ valid in the tangent space of $\mathbb{B}_{c_t}^n$ is mathematically ill-defined in $\mathbb{B}_{c_{t+1}}^n$ without a complex "curvature transport" operation, which is computationally prohibitive.

- **Adam's effectiveness:** By performing optimization in the projected coordinate space (treating the parameters as Euclidean vectors for the update step, then projecting back), Adam bypasses the need to track the changing geometric curvature explicitly in the momentum buffer. Empirically, this proved more stable for our mixed-curvature setting.

### B.7. Theoretical Analysis of Our Algorithm

In this section, we provide a error bound for our algorithm. It corresponds to proving that our procedure minimizes a target error bound (Redko et al., 2017). We first give the definition of error of a hypothesis $h$ on a domain $d$,

**Definition B.1.** The error of a hypothesis function $h \in \mathcal{H}$ with respect to a labeling function $f$ on a domain $d$ is given by

$$\epsilon_d(h, f) = \mathbb{E}_{(u,i) \in d}[|h(u, i) - f(u, i)|]. \tag{25}$$

Given a source domain equipped with a ground labeling function $(\mathcal{D}_s, f_s)$, we will adopt the abbreviation $\epsilon_s(h) = \epsilon_{d_s}(h, f_s)$. Similarly, the target domain can be denoted as $\epsilon_t(h) = \epsilon_{d_t}(h, f_t)$.

**Lemma B.2.** *Let $\mu_s, \mu_t \in \mathcal{P}(\mathcal{X})$ be two probability measures on $\mathcal{X} \subset \mathbb{R}^d$. Assume that the cost function $c(x, y) = \|\phi(x) - \phi(y)\|_{\mathcal{H}_{k_\epsilon}}$, where $\mathcal{H}_{k_\epsilon}$ is a Reproducing Kernel Hilbert Space (RKHS) with associated kernel $k_l \colon \mathcal{X} \times \mathcal{X} \to \mathbf{R}$ induced by $\phi : \mathcal{X} \to \mathcal{H}_{k_\epsilon}$ and $k_l(x, y) = \langle \phi(x), \phi(y) \rangle \mathcal{H}_{k_\epsilon}$. Assume further that the loss function $l_{h,f} \colon x \to l(h(x), f(x))$ is convex, symmetric, bounded, obeys the triangle inequality and has the parametric form $|h(x) - f(x)|^q$ for some $q > 0$. Assume also that $k_1$ is square-root integrable w.r.t. both $\mu_s$ and $\mu_t$, for all $\mu_s, \mu_t \in \mathcal{P}_p(\mathcal{X})$, where $\mathcal{X}$ is separable and $0 \leq k_{l(x,y)} \leq K, \forall x, y \in \mathcal{X}$. Then, the following holds,*

$$\epsilon_t(h, h') \leq \epsilon_s(h, h') + W_1(\mu_s, \mu_t), \tag{26}$$

*for every hypothesis $h, h' \in \mathcal{H}_{k_\epsilon}$.*

**Theorem B.3.** *Let $\mu_s, \mu_t \in \mathcal{P}(\mathcal{U}, \mathcal{V})$ be two joint probability measures on $\mathcal{U} \times \mathcal{V} \subset \mathbb{R}^{2 \times d}$, and $\mathcal{U}$ and $\mathcal{V}$ are independent. Assume that the cost function $c((u_i, v_i), (u_j, v_j)) = \|\phi(u_i, v_i) - \phi(u_j, v_j)\|_{\mathcal{H}_{k_\epsilon}}$. Under the assumption in Lemma B.2 for any $h, h' \in \mathcal{H}$, the following inequality holds,*

$$\epsilon_t(h, h') \leq \epsilon_s(h, h') + W_1(\mu_{s^u}, \mu_{t^u}) + \tau_1, \tag{27}$$

*where $\mu_{s^u}, \mu_{t^u} \in \mathcal{P}(\mathcal{U})$ be two probability measures on $\mathcal{U} \subset \mathbb{R}^d$, $\tau_1 = W_1(\mu_{s^v}, \mu_{t^v})$ and $\mu_{s^v}, \mu_{t^v} \in \mathcal{P}(\mathcal{V})$ be two probability measures on $\mathcal{V} \subset \mathbb{R}^d$.*

*Proof.* According to Lemma B.2,

$$\begin{aligned}
\epsilon_t(h, h') &\leq \epsilon_s(h, h') + W_1(\mu_s, \mu_t) \\
&= \epsilon_s(h, h') + W_1((\mu_{s^u}, \mu_{s^v}), (\mu_{t^u}, \mu_{t^v})).
\end{aligned} \tag{28}$$

Using the definition of Wasserstein distance, we have

$$\begin{aligned}
&W_1((\mu_{s^u}, \mu_{s^v}), (\mu_{t^u}, \mu_{t^v})) \\
&= \inf_{\gamma \in \prod(\mu_s, \mu_t)} \int_\Omega c((u_i, v_i), (u_j, v_j)) d\gamma((u_i, v_i), (u_j, v_j)) \\
&\leq \inf_{\gamma \in \prod(\mu_{s^u}, \mu_{t^u})} \int_\Omega c(u_i, u_j) d\gamma(u_i, u_j) \\
&\quad + \inf_{\gamma \in \prod(\mu_{s^v}, \mu_{t^v})} \int_\Omega c(v_i, v_j) d\gamma(v_i, v_j) \\
&= W_1(\mu_{s^u}, \mu_{t^u}) + W_1(\mu_{s^v}, \mu_{t^v}).
\end{aligned} \tag{29}$$

So we can conclude the proof. □

In RSs, there are some overlapping users in all domains. So $W_1(\mu_{s^u}, \mu_{t^u})$ can be decomposed two part: $W_1^{overlap}(\mu_{s^u}, \mu_{t^u})$ and $W_1^{non-overlap}(\mu_{s^u}, \mu_{t^u})$, and we treat each user as an independent distribution based on its transaction with items. Therefore,

$$W_1(\mu_{s^u}, \mu_{t^u}) \leq \sum_{i=1}^{|U_{overlap}|} W_1(\mu_{s_i}, \mu_{t_i}) + \tau_2, \tag{30}$$

where $|U_{overlap}|$ is the number of overlapping users, and $\tau_2 = W_1^{non-overlap}(\mu_{s^u}, \mu_{t^u})$. Thus, the result given in Theorem B.3 can be written as

$$\epsilon_t(h, h') \leq \epsilon_s(h, h') + \sum_{i=1}^{|U_{overlap}|} W_1(\mu_{s_i}, \mu_{t_i}) + \tau_1 + \tau_2. \tag{31}$$

The definition of error can be naturally extended to the case of many source domains. For each $u$, given a vector $\alpha^u \in \mathbb{R}^M$ such that $\sum_{j=1}^M \alpha_j^u = 1$. The $\alpha$-weighted error of $h$ is given by

$$\epsilon_\alpha(h) = \sum_{j=1}^M \alpha_j \epsilon_{d_{s_j}}(h). \tag{32}$$

Notice that Eq. (32) can be approximated by substituting the expectation operator by an empirical mean. In this case, we can denote the empirical error functional by

$$\hat{\epsilon}_\alpha(h) = \sum_{j=1}^{M} \frac{\alpha_j}{n_{s_j}} \sum_{i=1}^{n_{s_j}} |h(u_i, v_i) - f_j(u_i, v_i)|. \tag{33}$$

Moreover, for the purposes of the next theorem, we will suppose that $n_{s_j} = \beta_j n$ for $\sum_{j=1}^{M} \beta_j = 1$.

**Lemma B.4.** *Let $\mu$ be a probability measure in $\mathbb{R}^d$, so that for some $\alpha > 0$ we have that $\int_{\mathbb{R}^d} e^{\alpha \|x\|^2} d\mu < \infty$ and $\hat{\mu} = \frac{1}{n} \sum_{i=1}^{n} \delta_{x_i}$ be its associated empirical measure defined on a sample of independent variables $\{x_i\}_{i=1}^{n}$ drawn from $\mu$. Then for any $d' > d$ and $\xi < \sqrt{2}$ there exists a constant $n_0$ depending on $d'$ and some square exponential moment of $\mu$ such that for any $\epsilon > 0$ and $n > n_0 \max(\epsilon^{-(d'+2)}, 1)$,*

$$P[W_1(\hat{\mu}, \mu) > \epsilon] \le \exp\left(-\frac{\xi'}{2} n \epsilon^2\right), \tag{34}$$

*where $d'$ and $\xi'$ can be calculated explicitly.*

The consequence of Lemma B.4 is that we may express $\epsilon$ in terms of $\delta$,

$$\epsilon = \sqrt{\frac{2}{n\xi'} \log(\frac{1}{\delta})}. \tag{35}$$

**Lemma B.5.** *Under the assumptions of Lemma B.4, let $X$ be a sample of size $n$, where for each $j = 1, \ldots, D$, $\beta_j n$ points are drawn from $\mu_{s_j}$ and labelled according to $f_j$. Then, for any fixed $\alpha$, with probability $1 - \delta$ for all $h$ the following holds,*

$$P\left[|\hat{\epsilon}_\alpha(h) - \epsilon_\alpha(h)| > \epsilon + \theta\right] \le 2 \exp\left(\frac{-\epsilon^2 n}{2K \sum_{j=1}^{M} \frac{\alpha_j^2}{\beta_j}}\right), \tag{36}$$

*where $\theta = 2\sqrt{K/n} \sum_{j=1}^{M} \frac{\alpha_j}{\beta_j n \sqrt{\beta_j}}$.*

The consequence of Lemma B.5 is that we may express $\epsilon$ in Eq. (36) as,

$$\epsilon = \sqrt{\frac{2K \sum_{j=1}^{M} \frac{\alpha_j^2}{\beta_j} \log(\frac{2}{\delta})}{n}}. \tag{37}$$

**Theorem B.6.** *Let $X_{s_j}, j = 1, \ldots, D$, $X_t$ be N+1 samples with size $n_{s_j}$ and $n_t$ be drown i.i.d from $\mu_{s_j}$ and $\mu_t$ respectively. Let $\hat{\mu}_{s_j}$ and $\hat{\mu}_t$ be the respective empirical measures. If $\hat{h}_\alpha$ is the empirical minimizer of $\hat{\epsilon}_\alpha$ and $h_t^* = \arg\min_{h \in \mathcal{H}} \epsilon_t(h)$, then for any fixed $\alpha$ and $\delta \in (0, 1)$, with probability at least $1 - \delta$ (over the choice of samples), we have*

$$\epsilon_t(\hat{h}_\alpha) \le \epsilon_t(h_t^*) + c_1 + 2 \sum_{j=1}^{M} \alpha_j \left(\sum_{i=1}^{|U_{overlap}|} W_1(\mu_{s_{j_i}}, \mu_{t_i}) + c_2\right) + \tau_1 + \tau_2 + \lambda_j), \tag{38}$$

*where*

$$c_1 = \sqrt{\frac{2K \sum_{j=1}^{M} \frac{\alpha_j^2}{\beta_j} \log(\frac{2}{\delta})}{n}} + 2\sqrt{\frac{K}{n}} \sum_{j=1}^{M} \frac{\alpha_j}{\beta_j n \sqrt{\beta_j}},$$

$$c_2 = \sqrt{\frac{2 \log(\frac{1}{\delta})}{\eta'}} \left(\sqrt{\frac{1}{n_{s_{j_i}}}} + \sqrt{\frac{1}{n_{t_i}}}\right)$$

$$\tau_1 = W_1(\mu_{s_j^v}, \mu_{t_j^v}),$$

$$\tau_2 = W_1^{non-overlap}(\mu_{s_j^u}, \mu_{t^u})$$

*and*

$$\lambda_j = \epsilon_{s_j}(h_j^*) + \epsilon_t(h_j^*).$$

**Proof.** According to Eq. (32), we have

$$|\epsilon_\alpha(h) - \epsilon_t(h)| = |\sum_{j=1}^{M} \alpha_j \epsilon_{s_j}(h) - \epsilon_t(h)| \le \sum_{j=1}^{M} \alpha_j |\epsilon_{s_j}(h) - \epsilon_t(h)|. \tag{39}$$

Let $h_j^* = \arg\min_{h \in \mathcal{H}} \epsilon_{s_j}(h) + \epsilon_t(h)$. Then we can rewrite $|\epsilon_{s_j}(h) - \epsilon_t(h)|$ as

$$\begin{aligned} |\epsilon_{s_j}(h) - \epsilon_t(h)| &= |\epsilon_{s_j}(h) - \epsilon_{s_j}(h, h_j^*) \\ &+ \epsilon_{s_j}(h, h_j^*) - \epsilon_t(h, h_j^*) \\ &+ \epsilon_t(h, h_j^*) - \epsilon_t(h)|, \end{aligned} \tag{40}$$

Then, the triangle inequality yields that

$$\begin{aligned} f|\epsilon_{s_j}(h) - \epsilon_t(h)| &= |\epsilon_{s_j}(h) - \epsilon_{s_j}(h, h_j^*)| \\ &+ |\epsilon_{s_j}(h, h_j^*) - \epsilon_t(h, h_j^*)| \\ &+ |\epsilon_t(h, h_j^*) - \epsilon_t(h), \end{aligned} \tag{41}$$

In the last equality, by using the triangle inequality again, we get the following bounds. i.e.,

$$\begin{aligned} |\epsilon_{s_j}(h) - \epsilon_{s_j}(h, h_j^*)| &= |\epsilon_{s_j}(h, f_{s_j}) - \epsilon_{s_j}(h, h_j^*)| \\ &\le |\epsilon_{s_j}(h_j^*)| = |\epsilon_{s_j}(h_j^*). \end{aligned} \tag{42}$$

Plugging back these results into Eq. (39), we have

$$|\epsilon_\alpha(h) - \epsilon_t(h)| \le \sum_{j=1}^{M} (\epsilon_{s_j}(h_j^*) + \epsilon_t(h_j^*) + |\epsilon_{s_j}(h, h_j^*) - \epsilon_t(h, h_j^*)|). \tag{43}$$

Notice that $\lambda_j = \epsilon_{s_j}(h_j^*) + \epsilon_t(h_j^*)$. Moreover, using Eq. (31), we get

$$\epsilon_t(h, h') \le \epsilon_s(h, h') + \sum_{i=1}^{|U_{overlap}|} W_1(\mu_{s_i}, \mu_{t_i}) + \tau_1 + \tau_2, \tag{44}$$

which yields

$$|\epsilon_\alpha(h) - \epsilon_t(h)| \le \sum_{j=1}^{M} (\lambda_j + \sum_{i=1}^{|U_{overlap}|} W_1(\mu_{s_{j_i}}, \mu_{t_i}) + \tau_1 + \tau_2) \tag{45}$$

and

$$\epsilon_t(h) \le \epsilon_\alpha(h) + \sum_{j=1}^{M} (\lambda_j + \sum_{i=1}^{|U_{overlap}|} W_1(\mu_{s_{j_i}}, \mu_{t_i}) + \tau_1 + \tau_2)). \tag{46}$$

The bound we want to prove is achieved by bounding different terms in this equation for $h = \hat{h}$. By the triangle inequality, we have

$$W_1(\mu_{s_{j_i}}, \mu_{t_i}) \le W_1(\mu_{s_{j_i}}, \hat{\mu}_{s_{j_i}}) + W_1(\hat{\mu}_{s_{j_i}}, \hat{\mu}_{t_i}) + W_1(\mu_{t_i}, \hat{\mu}_{t_i}). \tag{47}$$

Now, we may bound each term $W_1(\mu, \hat{\mu})$ by Lemma 2, especially through Eq. (35)

$$
\begin{aligned}
W_1(\mu_{s_{j_i}}, \hat{\mu}_{s_{j_i}}) &\leq \sqrt{\frac{2\log(\frac{1}{\delta})}{\eta'}}\sqrt{\frac{1}{n_{s_{j_i}}}}, \\
W_1(\mu_{t_i}, \hat{\mu}_{t_i}) &\leq \sqrt{\frac{2\log(\frac{1}{\delta})}{\eta'}}\sqrt{\frac{1}{n_{t_i}}},
\end{aligned}
\tag{48}
$$

which ultimately leads to

$$
W_1(\mu_{s_{j_i}}, \mu_{t_i}) \leq W_1(\hat{\mu}_{s_{j_i}}, \hat{\mu}_{t_i}) + \sqrt{\frac{2\log(\frac{1}{\delta})}{\eta'}}\left(\sqrt{\frac{1}{n_{s_{j_i}}}} + \sqrt{\frac{1}{n_{t_i}}}\right).
\tag{49}
$$

The second term on the right-hand-side of Eq. (49) is exactly $c_2$, for which we get by bounding the Wasserstein distance in Eq. (46). Now, following the proof of Theorem 4 in (Ben-David et al., 2010), combining Lemma 3 we get,

$$
\epsilon_\alpha(\hat{h}_\alpha) \leq \hat{\epsilon}_\alpha(\hat{h}_\alpha) + \sqrt{\frac{2K\sum_{j=1}^M \frac{\alpha_j^2}{\beta_j}\log(\frac{2}{\delta})}{n}} + 2\sqrt{\frac{K}{n}}\sum_{j=1}^M \frac{\alpha_j}{\beta_j n\sqrt{\beta_j}},
\tag{50}
$$

Combining Eqs. (46), (49) and (50), we can get

$$
\epsilon_t(\hat{h}_\alpha) \leq \epsilon_t(h_t^*) + c_1 + 2\sum_{j=1}^M \alpha_j\left(\sum_{i=1}^{|U_{overlap}|} W_1(\mu_{s_{j_i}}, \mu_{t_i}) + \tau_1 + \tau_2 + \lambda_j + c_2\right).
\tag{51}
$$

So we can conclude the proof.

Theorem B.6 demonstrates that by minimizing the Wasserstein distance on the right-hand side of Eq. (38), we can effectively minimize the upper bound of error. Furthermore, it is equally important to minimize both $\tau_2$ and $\lambda_j$. $\tau_2$ represents the Wasserstein distance of the representation distributions between non-overlapping users across different domains; $\lambda_j$ reflects the model's inherent prediction bias. To first minimize the Wasserstein distance of overlapping users, we introduce a disentanglement encoder and a Wasserstein barycenter strategy. To minimize $\tau_2$, we employ a pseudo-label strategy to facilitate the representation learning of non-overlapping users, aiming to quickly reduce the Wasserstein distance of the representation distributions of non-overlapping users. Finally, to minimize $\lambda_j$, we implement this through the decoder reconstruction loss and an additional Binary Relevance Pairing (BRP) loss, which mainly focuses on maximizing the consistency between the model's predictions and the actual labels.

## C. Experimental Supplementation

To evaluate the effectiveness of HVAE in real-world online scenarios, we conduct an online A/B test on the Ant Group recommendation platform. Table.3 presents the statistical distribution of data across five real-world scenario domains, with a cross-domain usage overlap ratio of merely $1\%$ among all domains.

*Table 3.* Statistics of the online dataset.

| Domain | A | B | C | D | E |
|---|---|---|---|---|---|
| #User | 8520783 | 9633664 | 524483 | 1484030 | 7178325 |
| #Item | 2007 | 1189 | 324 | 75 | 1152 |
| #Avg.I/User | 3627.30 | 8007.02 | 635.03 | 5257.04 | 7560.69 |
| #Avg.U/Item | 1.46 | 1.80 | 1.10 | 1.41 | 1.45 |
| #Actions | 7279996 | 9520348 | 205751 | 394278 | 8709912 |
| #Sparsity | 99.96% | 99.92% | 99.88% | 99.65% | 99.89% |

