# OpenReview forum: "HVAE: Hyperbolic Variational Autoencoder For Flexible Knowledge Transfer Across Multiple Domains"
_ICML.cc/2026/Conference — ICML 2026 regular_

### Official Review · Reviewer_12gL · 2026-03-07

**Soundness:** 3
**Presentation:** 3
**Significance:** 2
**Originality:** 3
**Overall Recommendation:** 4
**Confidence:** 4

**Summary:**

To address the issue of geometric distribution mismatch in cross-domain recommendation task, this paper proposes a model that natively operates in hyperbolic space, thereby effectively solving the problems of cold start and data sparsity.

**Compliance With Llm Reviewing Policy:**

Affirmed.

**Final Justification:**

Thanks for the authors for their response to my comments. I found that my concerns have been well addressed.

**Key Questions For Authors:**

Please see the Weaknesses.

**Strengths And Weaknesses:**

Strengths

1. The theoretical analysis and experimental verification in the paper are detailed. The proposed HVAE model outperforms the selected baselines in large-scale industrial and public datasets.

2. The HVAE model effectively utilizes the property of hyperbolic space to handle data in long-tail scenarios.

3. Extensive ablation experiments effectively verifies the contribution of each component.

Weaknesses

1. The comparison experiments lack baselines from recent years. Comparing the model with more recent baselines, e.g., [a] and [b], would better demonstrate the advantage of HVAE.

2. Experimental data on model efficiency and parameter sensitivity of weights for different loss terms are lacking. Adding related experiments will help verify the overall performance of HVAE and understand the contribution of different loss terms.

3. It would be better to directly mention the content of “the Scalability and Robustness” in the abstract.

4. Many formulas contain missing or incorrect punctuation marks, e.g., Eq. (9), (10), and (11). The entire paper needs further review.

[a] Li, Hourun, et al. "DisCo: graph-based disentangled contrastive learning for cold-start cross-domain recommendation." Proceedings of the AAAI Conference on Artificial Intelligence. Vol. 39. No. 11. 2025.

[b] An, Zhicheng, et al. "DDCDR: A disentangle-based distillation framework for cross-domain recommendation." Proceedings of the 30th ACM SIGKDD Conference on Knowledge Discovery and Data Mining. 2024.

---

> ### Author Rebuttal · Authors · 2026-03-29
>
> **QA-1、Response to Key Question 1**
>
> We sincerely thank the reviewer for highlighting these two recent and highly relevant baselines. We completely agree that both DisCo and DDCDR represent excellent advancements in cross-domain recommendation. The primary reason they were not initially included in our main experiments is a fundamental architectural divergence: DisCo and DDCDR are natively designed for dual-domain (bipartite) knowledge transfer, whereas HVAE is explicitly architected to solve the multi-domain ($N \ge 3$) transfer problem under extreme sparsity constraints.
>
> To rigorously benchmark them in our multi-domain configuration, we would need to systematically adapt their bipartite architectures. However, extending DDCDR's adversarial distillation or DisCo's intent-wise contrastive graphs to $N$ domains necessitates $C_N^2$ domain-domain pairwise alignment modules. This incurs prohibitive computational overhead and exacerbates optimization complexity and instability(especially since we would perform this extension in the absence of their open-sourced code). This is why we evaluated our model against widely validated dual-domain models (GA-DTCDR, CCDR) that we meticulously extended, alongside native multi-domain frameworks(GA-MTCDR, HeroGraph). As our manuscript demonstrates, natively multi-domain frameworks surpass extended bipartite baselines in almost all cases.
>
> To address your concern, we will have a detailed discussion on DisCo and DDCDR in our Related Work. Because our paper inherently focuses on multi-domain scalability, we respectfully suggest that this dual-domain comparison would be more appropriate for a separate, dedicated study on carefully created dual-domain sparsity and cold-start datasets.
>
> **QA-2、Response to Key Question 2**
>
> We thank the reviewer for the constructive suggestions. First, to provide a rigorous efficiency evaluation, we additionally conducted time complexity experiments on the MYbank dataset (30% overlap ratio) using a single NVIDIA A100 GPU. The parameter counts for strong baselines, e.g., LightGCN (23.8M), GA-MTCDR (25.02M), and HeroGraph (24.62M), which are strictly in the same magnitude as HVAE (29.13M). The average training and evaluation times for processing 256 samples of per batch are 8.06s/1.38s (LightGCN), 8.77s/1.83s (GA-MTCDR), 9.11s/1.54s (HeroGraph), and 11.29s/2.64s for HVAE. This demonstrates that HVAE achieves significant performance gains with a marginal, acceptable increase in training time overhead. Furthermore, our successful 7-day A/B test on Ant Group's live system validates that HVAE strictly meets real-time industrial inference constraints, even with 1% domain overlap and user bases approaching ten million.
>
> Second, regarding parameter sensitivity, we explicitly grid-searched the loss weights $\alpha_1$, $\alpha_2$, and $\alpha_3$ within ($\{10^{-5}, 10^{-4}, 10^{-3}, 10^{-2}, 10^{-1}, 1\}$) as noted in the Parameter Settings section. Besides, we additionally conducted a sensitivity experiments. For $\alpha_1$（Please refer to the https://anonymous.4open.science/api/repo/anon-research-2026-166E/file/sensitivity_curve.png?v=5b642995 link to view the plot of the sensitivity curves), HVAE's performance remains stable with only minor fluctuations across a wide range of values ($10^{-5}$ to $10^{-1}$), but degrades at extreme large values (e.g., $\ge 1.0$). This is because large values would heavily penalize the latent norm, pulling embeddings toward the origin, which neutralizes the Poincaré manifold's exponential volume advantage and inducing the Euclidean "crowding problem".
>
> The roles of the distance factor ($\alpha_2$) and discriminator factor ($\alpha_3$) are fundamentally structural. Their parameter sensitivity is similar to that of $\alpha_1$, but the mathematical logic behind the decline in performance at larger values is distinct. For $\alpha_2$, large values would over-penalize the Wasserstein distance, pushing embeddings toward the manifold's absolute boundary and causing severe numerical instability. A large $\alpha_3$ would trigger adversarial training collapse, as the encoder exploits a trivial shortcut, forcing domain-invariant features to be perfectly identical but stripped of practical representation utility. As large values for $\alpha_2$ and $\alpha_3$ predictably trigger geometric instability and adversarial collapse, we evaluated their core contributions primarily through binary ablation studies (w/o-Distance and w/o-Classify) in the main text. We will include the above sensitivity analyses in the revised Appendix.
>
> **QA-3、Response to Key Question 3**
>
> We appreciate this suggestion and will highlight HVAE's robustness and scalability in large real industrial scenarios in the abstract.
>
> **QA-4、Response to Key Question 4**
>
> We will fix the typographical errors in Eq. (9), (10), and (11), and completely proofread the paper.

---

### Official Review · Reviewer_McUh · 2026-03-08

**Soundness:** 4
**Presentation:** 4
**Significance:** 3
**Originality:** 4
**Overall Recommendation:** 4
**Confidence:** 3

**Summary:**

The paper presents HVAE, a hyperbolic variational autoencoder framework for cross domain recommendation. It argues that user item interactions often follow long tail and hierarchical patterns, and that hyperbolic space can represent such structure better than Euclidean space. HVAE separates user representations into invariant and domain specific parts, then aligns the invariant part across domains using a hyperbolic Wasserstein barycenter idea. A cycle based pseudo labeling strategy is used to help in very sparse or cold start settings. Results on Amazon and an industrial MYbank dataset show consistent gains, especially when user overlap is very small.

**Compliance With Llm Reviewing Policy:**

Affirmed.

**Key Questions For Authors:**

1. How accurate is the Wasserstein approximation compared with a sampling based estimate across different curvatures and variances?
2. What prevents curvature learning from drifting to extreme values, and how sensitive is it to initialization?
3. What is the added compute cost over strong baselines, both in training and inference, especially in the industrial setup?

**Limitations:**

The approach adds modeling and tuning complexity, including curvature learning, distribution alignment, and pseudo labeling. It is not yet clear how robust the gains are across other types of domain shift or other feedback settings beyond the reported datasets.

**Strengths And Weaknesses:**

pros:

1.	The motivation is clear and matches the long tail structure common in recommendation data.
2.	The framework is end to end and keeps both representation learning and transfer inside hyperbolic space, including adaptive curvature.
3.	The empirical results are strong in the hardest low overlap regime, and the paper includes an industrial online evaluation.
4.	Presentation is strong overall. The paper is clearly structured, provides a clear high level overview of the full framework, and includes ablation results that make the contributions easy to follow.

cons

1.	The proposed framework combines several tightly coupled components, including hyperbolic representation learning, cross domain alignment, and pseudo labeling, which increases engineering and tuning burden.
2.	The submission does not provide an anonymous public code repository link, which makes it harder to verify the results and adopt the method. Releasing code and configs would significantly improve reproducibility.

---

> ### Author Rebuttal · Authors · 2026-03-29
>
> **QA-1、Response to Key Question 1**
>
> We thank the reviewer for this insightful question. We address the impact of varying curvatures and variances together, as their interplay critically determines the accuracy and stability of optimal transport in hyperbolic space. For the Wrapped Normal distribution, the variance is intrinsically parameterized within the local Euclidean tangent space before being projected onto the manifold (Appendix B.4). Consequently, our approximation strategy—decoupling the mean distance via exact manifold geodesics and the variance discrepancy within the local tangent space—is geometrically faithful to the distribution's generative definition. Crucially, this strategy remains highly robust across curvature and variance scales where sampling-based estimates fundamentally fail. In highly negative curvature, the exponential volume expansion causes sampling-based Wasserstein estimates to suffer from extreme variance and gradient instability due to the conformal factor. By computing variance updates directly in the tangent space, our approach bypasses this conformal scaling, guaranteeing reasonable gradients even in extremely high-curvature regimes. In summary, our proposed strategy provides a consistently accurate and numerically reliable estimation across variance and curvature scales.
>
> **QA-2、Response to Key Question 2**
>
> We sincerely thank the reviewer for this insightful question. We prevent extreme curvature drift and ensure robust numerical stability through three interconnected mechanisms. 1、We initialize the adaptive curvature $c_d$ to a small scalar (e.g., 0.1). A large initial $c_d$ overly restricts the Poincaré ball, forcing unaligned embeddings near the boundary and triggering numerical instability (e.g., NaNs and gradient explosion) as the conformal factor approaches infinity before any meaningful gradient updates can occur. Conversely, a small-scale initialization provides a relaxed initial geometry, allowing embeddings to safely align and preventing premature drift into unstable numerical extremes at the very beginning of training. 2、During dynamic learning, our analytical approximation intrinsically bounds gradient norms.  Even in highly negative curvature regimes, while embeddings approaching the boundary in such extreme volume expansion regimes typically exacerbate gradient explosion, our analytical approximation naturally mitigates this risk by decoupling the transport cost. We evaluate the mean distance via manifold geodesics while computing the variance discrepancy strictly within the local tangent space. This localized operation effectively bounds the gradient norms associated with variance updates, preventing numerical overflow; 3、We utilize Adam rather than Riemannian Adam (Radam). Because Radam requires complex parallel transport of momentum vectors, it becomes mathematically unstable without computationally expensive and numerically unstable curvature transport when $c_d$ dynamically changes. By applying Adam in the tangent space followed by a strict manifold projection to guarantee $||x|| < 1/\sqrt{c} - \epsilon$, Adam bypasses the momentum buffer explosion that plagues hyperbolic models in highly negative curvature regimes, ensuring our VAE reconstruction remains consistently robust.
>
> **QA-3、Response to Key Question 3 and Weakness 1**
>
> We sincerely thank the reviewer for raising these practical deployment considerations. As discussed in QA-1 and QA-2, the training process of our framework is quite stable. To provide a rigorous efficiency evaluation, we additionally conducted time complexity experiments on the MYbank dataset (30% overlap ratio) using a single NVIDIA A100 GPU. The parameter counts for strong baselines, e.g., LightGCN (23.8M), GA-MTCDR (25.02M), and HeroGraph (24.62M), which are strictly in the same magnitude as HVAE (29.13M). The average training and inference times for processing 256 samples of per batch are 8.06s/1.38s (LightGCN), 8.77s/1.83s (GA-MTCDR), 9.11s/1.54s (HeroGraph), and 11.29s/2.64s for HVAE. This demonstrates that HVAE achieves significant performance gains with a marginal, acceptable increase in training time overhead. During inference, our successful 7-day A/B test (Section4.3) on Ant Group's live system validates that HVAE strictly meets real-time industrial inference constraints, even with multiple domains (at least five), 1% domain overlap and user bases approaching ten million.
>
> **QA-4、Response to Weakness 2**
>
> We strongly agree on the importance of reproducibility and are eager to open-source our work. The absence of a repository in our initial submission is solely due to the strict intellectual property approval processes mandated by our institute. We are actively expediting these internal legal workflows and firmly commit to releasing the complete code, hyperparameter configurations, and datasets to ensure our results are fully reproducible for the community.

---

> > ### Author Rebuttal · Reviewer_McUh · 2026-04-05
> >
> > Thanks for the authors for their response to my comments. I found that my concerns have been well addressed.

---

### Official Review · Reviewer_rCWx · 2026-03-13

**Soundness:** 3
**Presentation:** 3
**Significance:** 3
**Originality:** 3
**Overall Recommendation:** 4
**Confidence:** 3

**Summary:**

This paper addresses a major problem in Cross-Domain Recommendation: how to give good suggestions when there is very little overlap in users between two platforms (the cold-start issue). The authors argue that current methods fail because they use standard flat geometry (Euclidean space), which cannot properly represent the complex, hierarchical nature of real-world user data. To solve this, they propose HVAE, a new model that operates entirely in hyperbolic space, a curved geometry better suited for tree-like data structures. Instead of just tweaking existing models, they rebuilt the core system to work naturally in this curved space. The main contributions are: 1. The model naturally captures the hierarchy in user preferences that flat models miss by working directly in hyperbolic space. 2. They developed a new, fast math trick to align data distributions across different domains without the heavy computing costs usually required for this type of geometry. 3. The method is backed by solid theory and performs exceptionally well in tests, especially when data is very scarce. Most importantly, it has already been tested in a real-world system and showed clear improvements over current methods.

**Compliance With Llm Reviewing Policy:**

Affirmed.

**Final Justification:**

I will keep "4: Weak accept" as my final recommendation with taking into account both the paper and the rebuttal addressing my main concerns to reinforce my assessment

**Key Questions For Authors:**

1. The paper proposes a novel approximation for the hyperbolic Wasserstein barycenter to enable tractable training of the VAE. While the theoretical bounds are provided, could you elaborate on the empirical impact of this approximation in high-curvature regimes? Specifically, did you observe any instability or degradation in reconstruction quality when the optimal curvature parameter learned during training resulted in highly negative curvature values?

2. In hyperbolic learning, the curvature parameter is often sensitive to initialization and can be difficult to optimize jointly with model weights. How sensitive is your model's convergence and final performance to the initialization of the curvature parameter? Did you employ a fixed curvature, a learnable per-layer curvature, or a global learnable curvature? Furthermore, did you observe any cases where the model collapsed to a Euclidean solution during training, effectively negating the hyperbolic benefits?

**Limitations:**

yes

**Strengths And Weaknesses:**

Strengths:
- The paper addresses a practically important problem in cross-domain recommendation. The successful deployment and positive A/B test results on the MYbank system provide encouraging evidence that the method can deliver value in real-world large-scale recommendation environments.
- The work goes beyond simply replacing Euclidean embeddings with hyperbolic ones. Instead, the authors redesign the VAE-based pipeline—including encoding, disentanglement, and distribution alignment—within hyperbolic space. This provides an interesting perspective on modeling hierarchical user–item relationships in cross-domain recommendation.
- The paper provides theoretical results which provides and help motivate the design of the proposed framework and offer insight into how hyperbolic geometry may influence representation capacity.
- The experimental results show consistent improvements over several baselines, especially in scenarios with very low user overlap and severe data sparsity. This setting is particularly important for cross-domain recommendation systems.

Weaknesses:
- Although the model incorporates adaptive curvature for different domains, the paper provides limited empirical analysis of how sensitive the performance is to the initialization of these curvature parameters or how they evolve during training. A more detailed ablation or visualization of curvature dynamics could help clarify how the model adapts to different domain structures.
- While the paper presents an interesting integration of hyperbolic embeddings with a VAE-based framework for cross-domain recommendation, some of the individual components (e.g., hyperbolic representation learning and VAE-based recommendation) have appeared in prior work. It would be helpful if the authors more clearly articulated which aspects of the method constitute the main algorithmic novelty beyond this integration.

---

> ### Author Rebuttal · Authors · 2026-03-29
>
> **QA-1、Response to Key Question 1**
>
> We thank the reviewer for the insightful question. Empirically, we did not observe instability or degradation in reconstruction quality when the model learns highly negative curvatures. While embeddings approaching the boundary in such extreme volume expansion regimes (corresponding to highly negative curvatures) typically exacerbate gradient explosion, our analytical approximation naturally mitigates this risk by decoupling the transport cost. We evaluate the mean distance via manifold geodesics while computing the variance discrepancy strictly within the local tangent space. This localized operation effectively bounds the gradient norms associated with variance updates, preventing numerical overflow.
>
> The architectural stability is further reinforced by our optimization strategy. As detailed in Appendix B.6, we utilize the Adam optimizer over Riemannian Adam (Radam). Radam requires complex parallel transport of momentum vectors, which becomes mathematically unstable without computationally expensive and numerically unstable curvature transport when $c_d$ dynamically changes. By contrast, we apply Adam in the tangent space followed by a strict manifold projection to guarantee $||x|| < 1/\sqrt{c} - \epsilon$. This fundamentally bypasses the momentum buffer explosion that limits hyperbolic models in highly negative curvature regimes, ensuring our VAE reconstruction remains consistently robust.
>
> **QA-2、Response to Key Question 2 and Weakness 1**
>
> We sincerely thank the reviewer's insightful feedbacks. We agree that the initialization sensitivity and evolution characteristics of the learnable curvature are crucial for understanding the model's behavior.
>
> In our framework, we do not employ fixed or per-layer curvatures; rather, we assign a domain-specific learnable adaptive curvature $c_d$ to each domain. This design allows the manifold to dynamically adapt to the intrinsic heterogeneous structure of specific domains. Regarding initialization, the model's convergence is highly robust, provided $c_d$ is initialized to a small scalar (e.g., 0.1). Initializing with excessively large values would impose a tightly restricted Poincaré ball radius on randomly initialized, unaligned embeddings at Step 0. This initial geometric mismatch forces the random vectors too close to or beyond the manifold boundary, triggering immediate NaNs and gradient explosion due to the conformal factor approaching infinity before any optimization can even occur. Conversely, small scalar initializations provide a relaxed initial geometry. After this safe initialization, if the optimal curvature parameter learned during the dynamic training process grows to highly negative values, our model would effectively safeguard against instability (as discussed in response to QA-1) .
>
> Besides, we did not observe Euclidean collapse (where $c \to 0$). The theoretical reason is the Geometry-Distribution Mismatch. The datasets inherently exhibit power-law distributions and underlying hierarchical structures. If the model attempts to collapse to a flat Euclidean space ($c \to 0$), the geometric distortion would cause a spike in our joint learning objective, specifically the VAE reconstruction and BPR ranking losses. Thus, the loss landscape naturally acts as a barrier, forcing the model to maintain a negative curvature to minimize reconstruction distortion.
>
> **QA-3、Response to Weakness 2**
>
> We thank the reviewer for the opportunity to clarify our algorithmic contributions. Our work extends fundamentally beyond a naive integration of hyperbolic embeddings into a VAE. Motivated by the long-tail and hierarchical data distributions in real-world scenarios, we identify the Euclidean geometry-distribution mismatch as a fundamental bottleneck in cross-domain recommendation (CDR). To address this, we redesign the entire multi-domain transfer mechanism natively on a Riemannian manifold, underpinned by formal theoretical guarantees. Notably, we introduce two main algorithmic contributions. First, to tackle the real challenge of scaling to multiple domains with minimal user overlap, we introduce a Hyperbolic Wasserstein Barycenter mechanism. This utilizes closed-form gyrovector operations to aggregate domain-invariant representations, ensuring the model scales effectively to multiple domains and remains robust even in extreme cold-start settings. Second, to make this hyperbolic optimal transport computationally tractable, we derive an analytical $O(n)$ approximation of the 2-Wasserstein distance. By mathematically decoupling the manifold mean distance and the local tangent space variance, we ensure industrial scalability without sacrificing essential metric properties. We will revise the Introduction section to make these core algorithmic contributions explicitly clear.

---

> > ### Author Rebuttal · Reviewer_rCWx · 2026-04-03
> >
> > Thanks for the detailed rebuttal. I will keep my positive score

---

### Decision · Program_Chairs · 2026-04-30

**Decision:**

Accept (regular)

**Comment:**

This paper presents a principled framework for cross-domain recommendation, referred to as HVAE, that unifies knowledge extraction and transfer across multiple domains within a hyperbolic manifold. It redesigns the entire VAE pipeline, including representation learning, disentanglement, and distribution alignment, in an end-to-end manner within hyperbolic space. Experiments on real-world datasets demonstrate it achieves superior performance in scenarios with long-tail distributions and minimal domain overlap. All reviewers agree that the paper addresses a practically important problem in cross-domain recommendation, the method is sound and well justified by experiments. Most of reviewers are satisfied by the author response, confirming that their concerns are resolved by the rebuttal. I do not think the current paper is super strong but it deserved to be presented at the conference.